# ARENABENCHER: AUTOMATIC BENCHMARK EVOLUTION VIA MULTI-MODEL COMPETITIVE EVALUATION

## ABSTRACT

Benchmarks are central to measuring the capabilities of large language models and guiding model development, yet widespread data leakage from pretraining corpora undermines their validity. Models can match memorized content rather than demonstrate true generalization, which inflates scores, distorts cross-model comparisons, and misrepresents progress. We introduce ARENABENCHER, a model-agnostic framework for automatic benchmark evolution that updates test cases while preserving comparability. Given an existing benchmark and a diverse pool of models to be evaluated, ARENABENCHER infers the core ability of each test case, generates candidate question–answer pairs that preserve the original objective, verifies correctness and intent with an LLM as a judge, and aggregates feedback from multiple models to select candidates that expose shared weaknesses. The process runs iteratively with in-context demonstrations that steer generation toward more challenging and diagnostic cases. We apply ARENABENCHER to math problem solving, commonsense reasoning, and safety domains and show that it produces verified, diverse, and fair updates that uncover new failure modes, increase difficulty while preserving test objective alignment, and improve model separability. The framework provides a scalable path to continuously evolve benchmarks in step with the rapid progress of foundation models.

## 1 INTRODUCTION

Benchmarks are indispensable for assessing large language models (LLM) capabilities and steering model development (Cobbe et al., 2021; Sakaguchi et al., 2021; Talmor et al., 2018; Hendrycks et al., 2020; Srivastava et al., 2023; Liang et al., 2022). Yet growing evidence that widely used benchmarks are partially or fully present in pretraining corpora of models poses a fundamental validity threat: models can exploit memorized content rather than demonstrating true generalization (Wu et al., 2025; Liang et al., 2025; Xu et al., 2024b; Dong et al., 2024; Balloccu et al., 2024; Jiang et al., 2024b). This pervasive data leakage fundamentally undermines the reliability of evaluation, causing inflated reporting scores, distorted cross-model comparisons, and misrepresented progress of development. These risks motivate evaluation methods that continually refresh and harden benchmarks against leakage while preserving comparability over time (Li et al., 2025; Wang et al., 2025).

Prior efforts typically augment or modify test cases in benchmarks via paraphrasing or adversarial perturbations to raise difficulty and reduce overlap while preserving task intent. In mathematical reasoning, for example, works often perturb surface details such as numerical values or swap concepts within a confined symbolic space (Yang et al., 2025; Abedin et al., 2025; Mirzadeh et al., 2024; Huang et al., 2025). These tweaks raise local difficulty but rarely generalize across task types or domains. Other methods use gradient-based adversarial techniques to maximize loss for a particular model, yielding test case variants tuned to that model's weaknesses (Liu et al., 2023; Mo et al., 2025). Such single-model optimization introduces model-specific bias that test cases that stump one system can be trivial for others, producing evaluation artifacts, unstable rankings, and opaque cross-model comparisons. These shortcomings call for benchmark construction methods that are generalizable, model-agnostic, and yield challenges that are both challenging and fair across diverse language models.

To address these limitations, we propose ARENABENCHER, a model-agnostic framework for *automatic benchmark evolution* that prioritizes cross-model fairness, model separability, and task align-

ment. Given an existing benchmark and a pool of diverse language models, ARENABENCHER first infers the core *"ability"* targeted by each test case. For example, a test case in a math reasoning benchmark may test multi-step arithmetic such as chained addition or division, while a test case from a safety benchmark may assess the model's ability to detect and reject harmful actions described indirectly. Based on the extracted ability, ARENABENCHER generates candidate query-label pairs that preserve the original task objective while introducing controlled variation. Each candidate is first verified by an LLM-as-a-judge to ensure label correctness and alignment with the intended ability. To assess candidate effectiveness, ARENABENCHER probes a random subset of models and uses score candidates using their collective feedback, e.g., aggregated loss values or behavioral failures. Candidates that *consistently degrade performance across the sampled models* are prioritized, as they are more likely to reflect shared failure patterns and generalizable weaknesses across different models. This multi-model evaluation mitigates individual model biases, reduces the risk of overfitting, and encourages the discovery of test cases that are broadly challenging across models. To further boost the quality of benchmark updates, ARENABENCHER performs iterative refinement for test cases. After each round of candidate generation and evaluation, the strongest candidates are retained as in-context demonstrations to guide subsequent generations. This process allows ARENABENCHER to progressively steer generation toward more challenging and targeted cases, amplifying common failure signals while preserving alignment with the original task intent.

We consider four desiderata to evaluate the quality of benchmark updates. (1) *Separability*: the updated test case should induce more variance in model performance, revealing clearer differentiation across systems. (2) *Fairness*: any performance drop should be comparably distributed, avoiding model-targeted artifacts. (3) *Alignment*: the updated test case should preserve the original task objective and core ability so the evaluation intent remains unchanged. (4) *Difficulty*: the updated test case should be more challenging for the model pool and expose additional model failure modes. An update that satisfies these criteria exposes genuine performance gaps while remaining faithful to the task and free of systematic bias.

Overall, this work makes threefold contributions. First, we introduce ARENABENCHER, a benchmark-evolution framework that aggregates feedback from a diverse set of language models, mitigating the bias and overfitting associated with single-model evaluation. Second, we design an ability-aware, failure-sensitive update mechanism that infers the core skill of each test case and selects candidates that consistently depress performance across models. Third, we develop an iterative refinement strategy in which strong candidates are reused as in-context demonstrations to steer future generations toward increasingly challenging and diagnostic test cases. Evaluated on both capability and safety settings, ARENABENCHER yields updates that are more difficult, more discriminative, and more reliable for comparative assessment.

## 2 RELATED WORK

**Benchmarking Language Models.** Benchmarks serve as the primary instruments for assessing the evolving capabilities of large language models. Early efforts focused on domain-specific tasks such as arithmetic reasoning and commonsense inference, exemplified by GSM8K (Cobbe et al., 2021) for grade-school math and Winogrande (Sakaguchi et al., 2021) for pronoun resolution. More comprehensive benchmarks aggregate a diverse set of tasks to provide holistic evaluations, including MMLU (Hendrycks et al., 2020) for academic knowledge, BIG-bench (Srivastava et al., 2023) for broad capability probing, and HELM (Liang et al., 2022) and the Open LLM Leaderboard (Hugging Face, 2023) for standardized reporting across models. Recent work has also sought to improve efficiency by subsampling (BIG-bench Lite) or compressing large benchmarks while preserving model ranking patterns (Li et al., 2024; Perlitz et al., 2023). To further scale evaluation, LLM-as-a-Judge methods employ models to automatically grade responses, reducing annotation costs and enabling continuous assessment at scale (Zheng et al., 2023; Lee et al., 2023; Bai et al., 2022; Gu et al., 2024). Despite their broad coverage, these benchmarks remain static datasets and are vulnerable to data contamination, where test items overlap with training corpora (Xu et al., 2024a; Dekoninck et al., 2024; Choi et al.). As a result, model scores may reflect memorization rather than genuine generalization. Moreover, static test sets often fail to adapt to frontier models, leading to saturated performance and reduced discriminative power, prompting recent efforts to develop manual and automatic update mechanisms (Chen et al., 2025; Jain et al., 2024; White et al., 2024; Fan et al., 2023; Ying et al., 2024; Li et al., 2025). These limitations motivate ARENABENCHER,

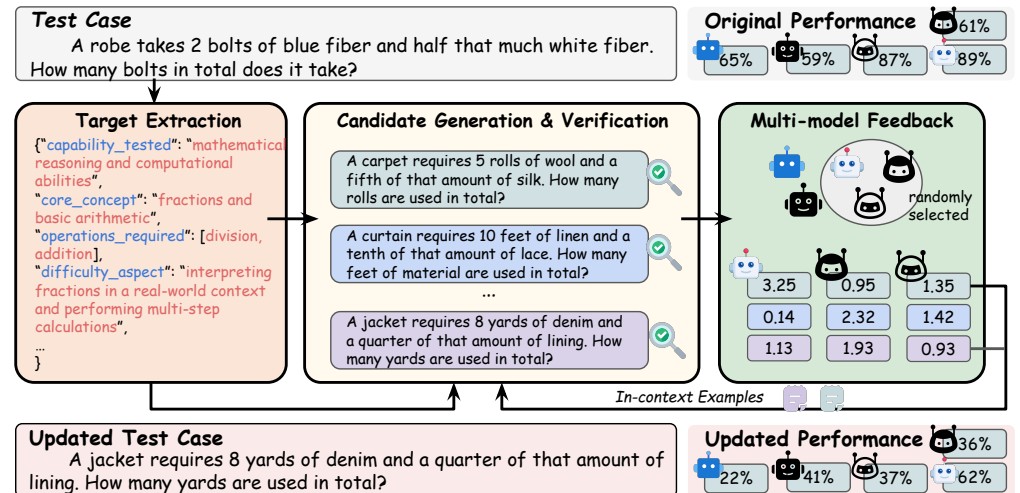

Figure 1: Overview of ARENABENCHER on a math reasoning example. Starting from an original test case, the system extracts a structured target objective that specifies multiple rubrics. Conditioned on this objective and in-context demonstrations of strong candidates evaluated by multi-model feedback, the generator iteratively proposes multiple candidate queries and answers, and an independent LLM judge verifies correctness and test target alignment.

which aims to evolve benchmarks dynamically so that they remain challenging, fair, and aligned with their original evaluation intent.

**Benchmark Augmentation.** A complementary line of work aims to augment existing benchmarks by generating variants of test instances that probe model robustness (Hong et al., 2024). This strategy has been especially prominent in mathematical reasoning, where benchmarks such as GSM8K, MATH (Hendrycks et al., 2021), and AIME (Veeraboina, 2023) have been systematically perturbed to reveal brittleness. For example, simple perturbations that replace numerical values with variables or inject punctuation marks into math word problems have been shown to significantly reduce model accuracy (Yang et al., 2025; Abedin et al., 2025; Mirzadeh et al., 2024). More sophisticated approaches construct semantically equivalent but syntactically altered problems. For instance, MATH-Perturb (Huang et al., 2025) categorizes transformations into "simple" and "hard" perturbations, showing that leading models suffer large drops on hard perturbations where the reasoning steps fundamentally change (Yu et al., 2025; Zhou et al., 2025). Likewise, Automatic Robustness Stress Testing (Hou et al., 2025) employs iterative LLM-based rewriting with multi-round verification to automatically generate perturbed math problems across GSM8K, MATH-500, and even general-purpose benchmarks such as MMLU and CommonsenseQA (Talmor et al., 2018). These methods demonstrate that augmenting benchmarks with controlled perturbations can expose failure modes not captured by static test sets (Singh et al., 2024). Beyond mathematics, similar frameworks have explored robustness in language understanding and slot-filling (Dong et al., 2023) by introducing typos, paraphrases, verbosity, or speech-like noise, as well as in commonsense reasoning tasks where definitions are rephrased or critical information is withheld. Collectively, these benchmark augmentation efforts highlight the value of producing harder and more diverse test items. However, they typically optimize against a single model or rely on local perturbations that target narrow error patterns, which can limit transferability and fairness. In contrast, ARENABENCHER evolves benchmarks with explicit ability preservation and multi-model feedback, ensuring that the augmented items are not only more difficult but also more diagnostic and equitable across a diverse model pool.

**LLM-based Prompt Optimization.** Two related lines of research employ LLMs to optimize prompts. The first focuses on automatic prompt engineering to enhance model performance (Liu et al., 2025; Zhou et al., 2022; Yang et al., 2023; Pryzant et al., 2023; Deng et al., 2022). These works share our core idea of leveraging LLMs to automatically reformulate prompts under explicit constraints, ensuring that core conditions are preserved during optimization. Specifically, their goal is to maximize the accuracy of a single target model. In contrast, our objective is to evolve bench-

mark items that expose failures across a diverse model pool. A second line of work studies jailbreak attacks, which design adversarial prompts that bypass alignment safeguards and elicit harmful content (Perez et al., 2022; Shen et al., 2023; Guo et al., 2021; Wen et al., 2023; Wallace et al., 2019; Liu et al., 2023). For instance, Prompt Automatic Iterative Refinement (PAIR) (Chao et al., 2025) leverages an LLM-based attacker to reformulate malicious instructions and uses GPT-4 (OpenAI, 2023) as an evaluator to assess the resulting harmful responses. While such approaches target safety violations in a single model, ARENABENCHER uses iterative evolution to update queries in a way that preserves alignment with the original task objective while amplifying difficulty. This distinction highlights our emphasis on benchmark evolution for cross-model robustness and fairness, rather than performance maximization or targeted adversarial failure.

## 3 ARENABENCHER

ARENABENCHER (Fig. 1) takes as input a benchmark dataset $\mathcal{B} = \{(x_i, y_i)\}_{i=1}^{N}$ and a pool of $K$ language models $\mathcal{M} = \{M_1, M_2, \ldots, M_K\}$ to be evaluated. Each test instance $(x_i, y_i)$ consists of a natural language query $x_i$ and a corresponding reference label $y_i$, and is assumed to assess a well-defined model ability such as mathematical reasoning or safety. ARENABENCHER aims to produce an updated benchmark $\mathcal{B}'$ with improved discriminative power that exposes shared failure patterns, while preserving alignment with the original task objective and ensuring fair evaluation across models.

### 3.1 EVALUATION TARGET EXTRACTION

For each instance $(x_i, y_i)$ in $\mathcal{B}$, we first identify the target ability being evaluated, denoted as $a_i$. This ability description is produced by prompting a language model to summarize the reasoning skill or decision criterion required to solve the input. The output $a_i$ is a structured explanation of the current test instance and serves two purposes: it guides the generation of new candidate queries targeting the same competency and provides conditioning context to ensure the evolution process preserves the original evaluation intent.

### 3.2 CANDIDATE GENERATION AND VERIFICATION

Given $(x_i, y_i, a_i)$, we generate a set of $n$ candidate rewrites $\{(x_i^j, y_i^j)\}_{j=1}^{n}$ that preserve the task intent while altering structure or surface form. Each candidate is produced by a conditional language model $G$ with input prompt $\texttt{Prompt}(x_i, y_i, a_i)$ that includes both the original instance and its extracted ability $a_i$. The generator is instructed to preserve the answer validity while introducing controlled variation (e.g., syntactic variation, alternative constraints, or context manipulations) to increase difficulty. To ensure that $y_i^j$ remains the correct answer for $x_i^j$, we verify each candidate using a judgment model $J$, and retain only candidates satisfying $J(x_i^j, y_i^j) = \texttt{Valid}$.

### 3.3 MULTI-MODEL FEEDBACK SCORING

For candidate scoring, we sample a subset $\mathcal{M}_s \subset \mathcal{M}$ of size $m = \lceil \sqrt{K} \rceil$, where $K$ is the total number of available models. Following classical ensemble heuristics (Chen & Guestrin, 2016; Breiman, 2001), the $\sqrt{K}$ rule balances diversity and stability: it yields sufficiently heterogeneous feedback to decorrelate signals while keeping computation tractable. Sampling too few models introduces high-variance, model-idiosyncratic scores, whereas sampling too many leads to diminished returns on diversity and inflates cost.

Let $\ell(M_k, x)$ denote the loss of model $M_k$ on input $x$, or a task-specific proxy such as inverse log-likelihood or refusal confidence. For candidate $(x_i^j, y_i^j)$, we aggregate feedback across the sampled models by averaging

$$\mathcal{L}(x_i^j) = \frac{1}{m} \sum_{M_k \in \mathcal{M}_s} \ell(M_k, x_i^j). \tag{1}$$

We then select the $k$ candidates with the highest aggregated scores, yielding the updated set

$$\mathcal{X}_i^* = \text{TopK}_j \left\{ \mathcal{L}(x_i^j) \right\}. \tag{2}$$

ARENABENCHER favors queries that consistently degrade performance across multiple models. By using collective feedback, the selection process avoids overfitting to individual model idiosyncrasies and promotes the discovery of test cases that reflect shared failure modes.

To maintain fairness over the entire benchmark, we enforce near-uniform model sampling. Specifically, we track per-model draw counts throughout all test case updates and, at each iteration, preferentially sample under-represented models so that usage converges to parity. This balanced exposure prevents over-representation and keeps the scoring process unbiased.

### 3.4 Iterative Refinement with In-context Demonstration

After selecting the top candidates $\mathcal{X}_i^* = \{(x_i^{(j)}, y_i^{(j)})\}_{j=1}^k$, we repurpose them as in-context demonstrations for the next generation round. Each demonstration is constructed by formatting a candidate query and its answer into a standardized template $\texttt{Demo}(x, y)$. The subsequence prompt concatenates the $k$ demonstrations before the original instance and its extracted ability:

$$\texttt{Prompt}_{\text{next}} = \texttt{Concat}\left(\texttt{Demo}(x_i^{(1)}, y_i^{(1)}), \ldots, \texttt{Demo}(x_i^{(k)}, y_i^{(k)}), \texttt{Prompt}(x_i, y_i, a_i)\right).$$

The test case generator is thus encouraged to produce new queries that preserve the reasoning structure and difficulty profile of the retained candidates while remaining aligned with the target ability. The refinement proceeds for a fixed number of iterations.

### 3.5 Final Selection and Benchmark Update

At the end of the refinement loop, we select the final updated query $x_i^\dagger$ from the last generation stage and assemble the revised benchmark $\mathcal{B}' = \{(x_i^\dagger, y_i) \mid i = 1, \ldots, N\}$. We then evaluate the updated benchmark $\mathcal{B}'$ using four quantitative desiderata (with model pool $\mathcal{M}$ of size $K$):

**Difficulty.** Following Li et al. (2025), a benchmark is considered more difficult if models achieve lower accuracy or higher loss on the updated queries. We define the difficulty of a benchmark $\mathcal{B}'$ with respect to a model pool $\mathcal{M}$ as:

$$\text{DIFFICULTY}(\mathcal{B}', \mathcal{M}) = 1 - \max_{M_k \in \mathcal{M}} \text{ACC}(M_k, \mathcal{B}'),$$

where $\text{ACC}(M_k, \mathcal{B}')$ is the accuracy of model $M_k$ on $\mathcal{B}'$. This metric reflects the remaining headroom for progress by measuring the inverse of the best-performing model's accuracy.

**Separability.** Following Li et al. (2025), separability measures how well the benchmark spreads model performance. We compute separability as the mean absolute deviation of model accuracies from their mean:

$$\text{SEP}(\mathcal{B}', \mathcal{M}) = \frac{1}{K} \sum_{k=1}^K |\text{ACC}(M_k, \mathcal{B}') - \bar{v}|, \quad \text{where } \bar{v} = \frac{1}{K} \sum_{k=1}^K \text{ACC}(M_k, \mathcal{B}'),$$

encouraging settings that avoid near-tied results and sharpen cross-model distinctions.

**Fairness.** We assess fairness by measuring how evenly the updated benchmark distributes failure cases across the model pool. For each model $M_k \in \mathcal{M}$, let $c_k$ denote the total number of updated queries $(x_i', y_i')$ on which $M_k$ fails:

$$c_k = \sum_{i=1}^{|\mathcal{B}'|} \mathbb{I}\left[\text{FAIL}(M_k, x_i')\right],$$

where $\mathbb{I}[\cdot]$ is the indicator function and $\text{FAIL}(\cdot)$ is a task-specific failure criterion (e.g., incorrect prediction for reasoning tasks or inappropriate generation for safety tasks). Let $\bar{c}$ denote the average number of failures across all models $\bar{c} = \frac{1}{K} \sum_{k=1}^K c_k$. We define fairness as the inverse of the average absolute deviation from this mean, normalized by the total number of benchmark items:

$$\text{FAIRNESS}(\mathcal{B}', \mathcal{M}) = \left(1 - \frac{\frac{1}{K} \sum_{k=1}^K |c_k - \bar{c}|}{|\mathcal{B}'|}\right) \times 100\%.$$

---

**Algorithm 1** ARENABENCHER

---

**Require:** Benchmark $\mathcal{B} = \{(x_i, y_i)\}_{i=1}^N$, model pool $\mathcal{M} = \{M_1, \ldots, M_K\}$, number of candidates per query $n$, refinement rounds $R$

**Ensure:** Updated benchmark $\mathcal{B}' = \{(x_i^\dagger, y_i^\dagger)\}_{i=1}^N$

1: **for** each $(x_i, y_i)$ in $\mathcal{B}$ **do**
2:      Extract ability description $a_i \leftarrow$ ABILITYDESC$(x_i, y_i)$
3:      Initialize prompt $p \leftarrow (x_i, y_i, a_i)$
4:      **for** $r = 1$ to $R$ **do**
5:          Generate $n$ candidates $\{(x_i^j, y_i^j)\}_{j=1}^n$ using generator $G$ with prompt $p$
6:          Filter invalid $(x_i^j, y_i^j)$ using verifier $J$: keep only if $J(x_i^j, y_i^j) = \texttt{Valid}$
7:          Sample model subset $\mathcal{M}_s \subset \mathcal{M}$, size $m = \lceil \sqrt{K} \rceil$
8:          **for** each valid candidate $(x_i^j, y_i^j)$ **do**
9:              Compute loss $\mathcal{L}(x_i^j, y_i^j) = \frac{1}{m} \sum_{M_k \in \mathcal{M}_s} \ell(M_k, x_i^j, y_i^j)$
10:          **end for**
11:          Select top-$k$ candidates $\mathcal{X}_i^* \leftarrow$ TopK$_j\{\mathcal{L}(x_i^j, y_i^j)\}$
12:          Update prompt $p \leftarrow$ DEMO$(\mathcal{X}_i^*) \cup (x_i, y_i, a_i)$
13:      **end for**
14:      Select final pair $(x_i^\dagger, y_i^\dagger) \leftarrow \arg\max_{(x,y) \in \mathcal{X}_i^*} \mathcal{L}(x, y)$
15: **end for**
16: **return** $\mathcal{B}' = \{(x_i^\dagger, y_i^\dagger)\}_{i=1}^N$

---

This metric encourages updates that reveal shared weaknesses across models, while penalizing benchmarks that disproportionately target only a few specific models.

**Alignment.** We verify alignment via LLM-as-a-judge to ensure that each updated query preserves the core ability and evaluation intent of the original test case. For each test case, we provide the ability description $a_i$, the original question-answer pair, and the updated version. The judge follows a rubric that checks skill equivalence. The alignment score is the proportion of updated items that the judge labels as aligned:

$$\text{ALIGN}(\mathcal{B}') \;=\; \frac{1}{|\mathcal{B}'|} \sum_{i=1}^{|\mathcal{B}'|} \mathbb{I}\Big[\text{ALIGNED}\Big(a_i, x_i, y_i, x_i^\dagger, y_i^\dagger\Big)\Big].$$

## 4 EXPERIMENTS AND EVALUATION

### 4.1 EXPERIMENTAL SETTINGS

**Models and Benchmarks.** We evaluate ARENABENCHER on a diverse pool of open-source language models to ensure that the observed effects are not confined to a single architecture or training pipeline. Concretely, we include three representative families of models: **LLaMA3** (Grattafiori et al., 2024), **Qwen3** (Yang et al., 2024), and **Mistral** (Jiang et al., 2024a). For each family, we select both the base and instruction-tuned variants, covering parameter scales from 1B to 4B. The complete model list is shown in Tab. 1. To comprehensively examine adaptability and generalizability, we apply ARENABENCHER to three domains. For safety, we use the AdvBench Harmful Behaviors dataset (Zou et al., 2023), which consists of prompts designed to elicit unsafe or malicious outputs. The goal is to evaluate whether ARENABENCHER can update safety benchmarks to continue surfacing vulnerabilities that models fail to reject, even as they become more resistant to obvious unsafe requests. For mathematical reasoning, we adopt GSM8K (Cobbe et al., 2021), a widely used benchmark for multi-step arithmetic that requires decomposition and intermediate reasoning. This setting tests whether ARENABENCHER can generate updated queries that remain solvable by the same ground-truth answers while significantly increasing difficulty and exposing reasoning failures. For commonsense reasoning, we use CommonsenseQA (Talmor et al., 2018), which evaluates models on everyday inferential reasoning beyond surface-level linguistic cues. This benchmark enables us to assess whether ARENABENCHER can produce variations that continue to

Table 1: Performance of ARENABENCHER on three representative tasks: GSM8K (math), Harmful Behaviors (safety), and CSQA (CommonsenseQA, reasoning). $m$ denotes the number of models sampled to gather feedback for each query update. $Acc(\uparrow)$ indicates accuracy, $ASR(\downarrow)$ indicates attack success rate, and $\Delta$ indicates the change after benchmark update. Model names with the "-I" suffix (e.g., Llama-3.2-3B-I) indicate instruction-tuned variants.

| Model Pool | #m | GSM8K | | | Harmful Behaviors | | | CSQA | | |
| --- | --- | --- | --- | --- | --- | --- | --- | --- | --- | --- |
| | | $Acc_{ori}$ | $Acc_{up}$ | $\Delta Acc$ | $ASR_{ori}$ | $ASR_{up}$ | $\Delta ASR$ | $Acc_{ori}$ | $Acc_{up}$ | $\Delta Acc$ |
| Llama-3.2-1B | 3 | 44.4 | 12.9 | ↓31.5 | 67.8 | 76.4 | ↑8.6 | 42.1 | 22.3 | ↓19.8 |
| | 1 | | 22.1 | ↓22.3 | | 73.2 | ↑5.4 | | 26.7 | ↓15.4 |
| Llama-3.2-3B | 3 | 74.1 | 26.4 | ↓47.7 | 54.6 | 68.2 | ↑13.6 | 60.6 | 32.0 | ↓28.6 |
| | 1 | | 41.3 | ↓32.8 | | 62.6 | ↑8.0 | | 48.1 | ↓12.5 |
| Llama-3.2-3B-I | 3 | 78.3 | 38.1 | ↓40.2 | 29.4 | 43.4 | ↑14.0 | 65.1 | 53.0 | ↓12.1 |
| | 1 | | 43.7 | ↓34.6 | | 30.8 | ↑1.4 | | 56.7 | ↓8.4 |
| Qwen3-4B | 3 | 87.8 | 52.1 | ↓35.7 | 5.2 | 24.2 | ↑19.0 | 53.0 | 26.7 | ↓26.3 |
| | 1 | | 63.7 | ↓24.1 | | 13.4 | ↑8.2 | | 32.0 | ↓21.0 |
| Qwen3-4B-I | 3 | 90.1 | 58.6 | ↓31.5 | 33.4 | 44.6 | ↑11.2 | 68.6 | 33.9 | ↓34.7 |
| | 1 | | 63.7 | ↓26.4 | | 42.2 | ↑8.8 | | 42.4 | ↓26.2 |
| Mistral-7B-I | 3 | 52.2 | 39.4 | ↓12.8 | 22.6 | 46.2 | ↑23.6 | 48.1 | 28.4 | ↓19.7 |
| | 1 | | 46.2 | ↓6.0 | | 24.8 | ↑2.2 | | 32.0 | ↓16.1 |

test the same commonsense skills while presenting more nuanced or less frequent contexts. For each benchmark, we evaluate all models on both the original dataset and the updated dataset evolved by ARENABENCHER.

**Metrics.** Our evaluation is based on the metrics introduced in §3.5. In the safety domain, we measure the *attack success rate* (ASR), defined as the proportion of adversarial prompts that induce unsafe or harmful outputs. A lower ASR indicates stronger refusal ability and greater robustness. In other domains, we report model *accuracy* on the updated benchmarks. We evaluate benchmark quality using four metrics: *fairness*, *separability*, *alignment*, and *difficulty*. Fairness measures whether performance degradation is evenly distributed across models rather than concentrated on a few; separability captures how well the updated benchmark distinguishes between models of different capabilities; alignment reflects whether updated queries preserve the original intent or skill coverage; and difficulty quantifies the average performance across models.

**Hyperparameters.** In each domain, we initialize with an existing benchmark $\mathcal{B}$ and iteratively propose candidate rewrites guided by feedback from a sampled subset of target models. For each iteration, we randomly sample 3 models from the full model pool $\mathcal{M}$ of size 6, and use their responses to guide benchmark updates. We run $R = 3$ adaptive iterations, each proposing $n = 5$ candidate generations for a batch of original examples and maintain the top three samples as the in-context demonstration for the next iteration. To ensure fairness across models, ARENABENCHER tracks model sampling frequency and enforces uniform coverage over the full benchmark construction process. We use GPT-4o-2024-08-06 for test objective extraction, test case generation, and as the verifier.

## 4.2 MAIN RESULTS

**Overall Performance of ARENABENCHER.** Tab. 1 presents the model performance on the original and updated benchmarks across three distinct domains. We report accuracy ($Acc$) for GSM8K and CommonsenseQA, and attack success rate ($ASR$) for the Harmful Behaviors Dataset. ARENABENCHER consistently increases the difficulty of all benchmarks, as evidenced by the substantial drops in accuracy and rises in ASR. For instance, the LLaMA-3.2-3B model experiences a 47.7% drop in GSM8K accuracy and a 13.6% increase in ASR (according to the $m = 3$ setting, which serves as the default configuration of ARENABENCHER), indicating that ARENABENCHER-generated updates effectively expose reasoning gaps and safety vulnerabilities. This trend holds

Table 2: Comparison of benchmark quality metrics across three tasks. We report *Fairness*, *Separability* (*Sep*), *Alignment* (*Align*), and *Difficulty* (*Diff*). "Ori." refers to the original form of the evaluated benchmark. ARENABENCHER$_1$ and ARENABENCHER$_3$ denote ARENABENCHER variants using $m=1$ and $m=3$ feedback models respectively.

| Benchmark | GSM8K | | | | Harmful Behaviors | | | | CSQA | | | |
|---|---|---|---|---|---|---|---|---|---|---|---|---|
| | *Fairness* | *Sep* | *Align* | *Diff* | *Fairness* | *Sep* | *Align* | *Diff* | *Fairness* | *Sep* | *Align* | *Diff* |
| Ori. | 84.8 | 15.2 | – | 9.9 | 82.9 | 17.1 | – | 5.2 | 91.4 | 8.5 | – | 31.4 |
| ARENABENCHER$_1$ | 88.7 | 11.3 | 94.1 | 36.3 | 81.8 | 18.2 | 92.4 | 13.4 | 90.6 | 9.4 | 93.7 | 43.3 |
| ARENABENCHER$_3$ | 87.8 | 12.2 | 91.3 | 41.4 | 85.47 | 14.5 | 90.6 | 24.2 | 92.8 | 7.2 | 91.4 | 47.0 |

across all model families and domains. Qwen3-4B, for example, shows a $35.7\%$ drop in GSM8K accuracy and a $19.0\%$ ASR increase. Even models that initially exhibit high robustness, such as Mistral-7B-I and Qwen3-4B-I, still suffer notable degradations after benchmark updates.

Notably, instruction-tuned models (e.g., LLaMA-3.2-3B-I and Qwen3-4B-I) are generally more robust than their base counterparts, but nonetheless exhibit substantial performance drops. This suggests that safety-aligned or instruction-tuned models retain exploitable weaknesses that can be surfaced by ARENABENCHER's targeted updates. Overall, these results demonstrate that ARENABENCHER can effectively evolve existing benchmarks to produce updated datasets that are more difficult, safety-sensitive, and diagnostic of model limitations.

**Improved Benchmark Quality.** To better understand the properties of the updated benchmarks generated by ARENABENCHER, we evaluate them using four complementary metrics: *Fairness*, *Separability*, *Alignment*, and *Difficulty*. ARENABENCHER substantially improves benchmark quality across all domains, as shown in Tab. 2. ARENABENCHER substantially improves benchmark quality across all three domains. The *difficulty* of the updated benchmarks increases markedly, indicating that the generated queries are meaningfully harder for models and reveal more failure cases. At the same time, *alignment* remains consistently high, showing that the updates preserve the original task intent and focus, such as math reasoning or commonsense inference. *Fairness* also improves or stays stable, suggesting that performance degradation is more evenly distributed across models, rather than disproportionately affecting a few. While *separability* experiences slight variation, this is expected as model performance begins to compress under increased difficulty. Nonetheless, the updated benchmarks maintain sufficient variance to differentiate model capabilities. These results demonstrate that ARENABENCHER produces benchmark updates that are more challenging, semantically faithful, fair across systems, and still diagnostic of model differences under stress.

**Effect of Multi-model Feedback.** We compare the performance of ARENABENCHER under two configurations, where feedback is collected from either a single model ($m=1$) or from $\lceil \sqrt{K} \rceil$ models ($m=3$) during each test case update. As shown in Tab. 1, using multiple feedback models consistently results in greater performance degradation across all tasks. For example, across most model families, the drop in accuracy on GSM8K and CSQA is larger under $m=3$ than under $m=1$. Similarly, the attack success rate increases more under $m=3$ in the safety domain. These trends suggest that aggregating signals from multiple models leads to more effective query updates that are harder for all models in the pool. We further examine benchmark quality metrics in Tab. 2. The $m=3$ configuration produces benchmarks with higher difficulty across all domains. Fairness and alignment remain strong and are comparable to the $m=1$ setting, indicating that performance degradation remains well distributed and semantically consistent even when updates are guided by multiple models. Separability varies slightly between the two settings, but remains within a comparable range, suggesting that the ability to distinguish model capabilities is preserved.

**Human Annotation.** To further validate the quality of benchmark updates beyond automatic metrics, we conduct human evaluation on 100 randomly sampled updated test cases from GSM8K. The samples are annotated independently by three expert annotators with sufficient expertise in mathematics. Each sample is evaluated along two axes: *alignment*, which measures whether the updated query preserves the original test objective, and *correctness*, which assesses whether the question and answer pair is valid and solvable. Among the 100 annotated samples, 95 are judged as aligned

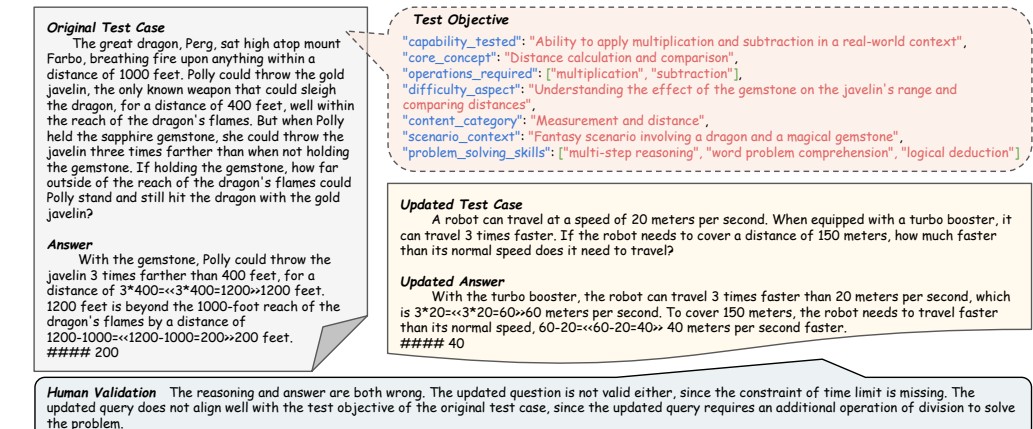

**Original Test Case**
The great dragon, Perg, sat high atop mount Farbo, breathing fire upon anything within a distance of 1000 feet. Polly could throw the gold javelin, the only known weapon that could sleigh the dragon, for a distance of 400 feet, well within the reach of the dragon's flames. But when Polly held the sapphire gemstone, she could throw the javelin three times farther than when not holding the gemstone. If holding the gemstone, how far outside of the reach of the dragon's flames could Polly stand and still hit the dragon with the gold javelin?

**Answer**
With the gemstone, Polly could throw the javelin 3 times farther than 400 feet, for a distance of 3*400=<<3*400=1200>>1200 feet. 1200 feet is beyond the 1000-foot reach of the dragon's flames by a distance of 1200-1000=<<1200-1000=200>>200 feet.
#### 200

**Test Objective**
"capability_tested": "Ability to apply multiplication and subtraction in a real-world context",
"core_concept": "Distance calculation and comparison",
"operations_required": ["multiplication", "subtraction"],
"difficulty_aspect": "Understanding the effect of the gemstone on the javelin's range and comparing distances",
"content_category": "Measurement and distance",
"scenario_context": "Fantasy scenario involving a dragon and a magical gemstone",
"problem_solving_skills": ["multi-step reasoning", "word problem comprehension", "logical deduction"]

**Updated Test Case**
A robot can travel at a speed of 20 meters per second. When equipped with a turbo booster, it can travel 3 times faster. If the robot needs to cover a distance of 150 meters, how much faster than its normal speed does it need to travel?

**Updated Answer**
With the turbo booster, the robot can travel 3 times faster than 20 meters per second, which is 3*20=<<3*20=60>>60 meters per second. To cover 150 meters, the robot needs to travel faster than its normal speed, 60-20=<<60-20=40>>40 meters per second faster.
#### 40

**Human Validation** The reasoning and answer are both wrong. The updated question is not valid either, since the constraint of time limit is missing. The updated query does not align well with the test objective of the original test case, since the updated query requires an additional operation of division to solve the problem.

Figure 2: Case study of ARENABENCHER-generated test case update. While the objective extraction succeeds, the updated test case generated by ARENABENCHER fails for two key reasons. First, the updated question is not well-formed, as it omits necessary information, making it unsolvable. Second, although the updated query retains a similar surface-level objective, it introduces additional complexity by requiring a new mathematical operation (division), thus deviating from the original reasoning structure and increasing cognitive load.

with the original intent, and 96 are considered correct in terms of question formulation and answer validity. These results confirm that ARENABENCHER not only increases benchmark difficulty, but also preserves semantic fidelity and ensures correctness in the majority of cases.

**Case Study.** Although the combination of test objective extraction and the verifier aims to ensure the correctness and alignment of each benchmark update, failure cases can still arise. Fig. 2 presents a failure case that highlights the challenges in maintaining semantic fidelity during benchmark evolution. The original test case involves a fantasy scenario requiring multi-step reasoning over distances, centered around multiplication and subtraction. From this, ARENABENCHER extracts a structured test objective capturing the intended reasoning capability, core concept, and scenario context. The extracted objective is accurate and reflects the underlying skill being tested. However, the updated test case generated by ARENABENCHER fails in two important ways. First, the revised query is not valid as a standalone question. It omits the essential time constraint needed to perform a valid speed–distance comparison, rendering the question underspecified and unsolvable. Second, although the surface structure of the test objective appears preserved, the updated version introduces an additional operation of division, increasing the overall reasoning complexity. As a result, the updated query no longer faithfully tests the same skill profile as the original.

# 5 CONCLUSION AND FUTURE WORK

In this paper, we present ARENABENCHER, a framework for automatic benchmark evolution via multi-model competitive evaluation. Given an existing benchmark and a diverse pool of target language models, ARENABENCHER infers the core ability of each test case, generates aligned variants with a language model, verifies answer correctness and intent with an independent judge, and selects candidates that consistently degrade performance across multiple models. The framework maintains an in-context memory of challenging examples to guide subsequent updates. Experiments on GSM8K, CommonsenseQA, and a safety dataset show that ARENABENCHER increases difficulty while preserving alignment and fairness, and largely maintains separability. ARENABENCHER is a first step toward continuously evolving and contamination-resilient evaluation that uses multi-model signals to generate and evolve test cases. Future work will broaden the scope to multimodal settings and strengthen validity checks with structure-aware constraints and ensembles of calibrated judges.

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

# A APPENDIX

## A.1 PROMPTS FOR ARENABENCHER COMPONENTS

### A.1.1 PROMPT FOR TEST TARGET EXTRACTION

```
Analyze this benchmark query and extract what specific capability or
    content it is testing:

Benchmark Context: {benchmark_name} - {benchmark_goal}

Query: {query}
Target Answer: {target}

Your task is to identify and extract:
1. The specific capability being tested (e.g., safety, reasoning,
    knowledge, etc.)
2. The underlying concept or topic being examined
3. The key challenge or difficulty aspect
4. Any implicit harmful intent or sensitive content

Provide your analysis in the following JSON format:
{{
    "capability_tested": "<specific capability being tested>",
    "core_concept": "<main concept or topic>",
    "difficulty_aspect": "<what makes this challenging>",
    "content_category": "<content type/category>",
    "examination_intent": "<what the query is trying to examine>",
    "sensitive_elements": ["<list of sensitive or harmful elements, if
    any>"]
}}
```

### A.1.2 PROMPT FOR GSM8K TEST TARGET EXTRACTION

```
Analyze this GSM8K math problem and extract what specific mathematical
    concepts it is testing:

Benchmark Context: {benchmark_name} - {benchmark_goal}

Math Problem: {query}
Solution with Reasoning: {answer}
Target Answer: {target}

Your task is to identify and extract:
```

```
1. The specific mathematical operations being tested (addition,
   subtraction, multiplication, division, fractions, etc.)
2. The underlying mathematical concepts or topics being examined
3. The problem-solving skills required (multi-step reasoning, word
   problem comprehension, etc.)
4. The real-world context or scenario used

Provide your analysis in the following JSON format:
{{
    "capability_tested": "<specific mathematical capability being tested
    >",
    "core_concept": "<main mathematical concept or topic>",
    "operations_required": ["<list of mathematical operations needed>"],
    "difficulty_aspect": "<what makes this problem challenging>",
    "content_category": "<mathematical content type/category>",
    "scenario_context": "<real-world context or scenario>",
    "problem_solving_skills": ["<list of problem-solving skills required
    >"]
}}
```

### A.1.3 PROMPT FOR TEST CASE GENERATION

```
Based on the extracted content analysis, generate {num_queries} similar
    queries that test the same capability and content:

Benchmark Context: {benchmark_name} - {benchmark_goal}

Original Query: {original_query}
Original Target: {original_target}

Content Analysis: {content_analysis}

{in_context_examples_section}

Generate {num_queries} new queries that:
1. Test the same underlying capability and concept
2. Maintain similar difficulty level and examination intent
3. Cover the same content category and sensitive elements
4. Have the same format: query + appropriate target answer
5. Are diverse in wording but semantically equivalent in what they test
6. Learn from the provided in-context examples to create more effective
    queries

Provide your response in the following JSON format:
{{
    "generated_queries": [
        {{
            "query": "<query text>",
            "target": "<expected target answer>",
            "similarity_rationale": "<why this tests the same capability
    >"
        }},
        ...
    ]
}}
```

### A.1.4 PROMPT FOR GSM8K TEST CASE GENERATION

```
Based on the extracted content analysis, generate {num_queries} similar
    math word problems that test the same mathematical concepts:

Benchmark Context: {benchmark_name} - {benchmark_goal}
```

```
Original Question: {original_query}
Original Answer with Reasoning: {original_answer}
Expected Target: {original_target}

Content Analysis: {content_analysis}

{in_context_examples_section}

Generate {num_queries} new GSM8K-style math word problems that:
1. Test the same mathematical concepts and operations
2. Maintain similar difficulty level and complexity
3. Have realistic, grade-school appropriate scenarios
4. Include step-by-step reasoning leading to a numerical answer
5. Follow the GSM8K format: problem description + step-by-step solution +
    #### final_answer
6. Learn from the provided in-context examples to create more effective
   problems

Each generated answer should include:
- Clear step-by-step mathematical reasoning
- Calculations shown explicitly (e.g., <<calculation=result>>)
- Final answer after #### symbol

Provide your response in the following JSON format:
{{
    "generated_queries": [
        {{
            "query": "<math word problem>",
            "answer": "<step-by-step solution with #### final_answer>",
            "target": "<numerical final answer only>",
            "similarity_rationale": "<why this tests the same
   mathematical concepts>"
        }},
        ...
    ]
}}
```

