# OpenReview forum: "ArenaBencher: Automatic Benchmark Evolution via Multi-Model Competitive Evaluation"
_ICLR.cc/2026/Conference — Submitted to ICLR 2026_

### Official Review · Reviewer_Mg4w · 2025-10-29

**Soundness:** 2
**Presentation:** 3
**Contribution:** 2
**Rating:** 4
**Confidence:** 5

**Summary:**

This paper proposes ARENABENCHER, a framework for automatic benchmark evolution using multi-model feedback. The idea is to iteratively generate new test cases while preserving the core ability of each item and selecting those that degrade performance across several models. The authors claim that the method improves difficulty, fairness, and separability of existing benchmarks such as GSM8K, CSQA, and harmful behavior datasets.

**Strengths:**

1. The paper tackles an important problem: benchmark contamination and stagnation under rapid LLM progress.

2. The multi-model feedback mechanism is conceptually interesting and potentially more robust than single-model adversarial rewriting.

3. The framework is clearly modular and described in a step-by-step manner.

**Weaknesses:**

1. Major claims are not empirically justified. Many statements in Sections 1–3 are presented as facts without direct supporting evidence. For example, the claim that multi-model feedback “mitigates bias” is not demonstrated beyond intuition.

2. Experimental definitions are unclear or inconsistent. Key metrics such as fairness are vaguely defined in prose but not rigorously formalized in Section 4. In Table 2, fairness is said to represent “how evenly the performance drop is distributed,” but if so, the original dataset should always be 100% fair by definition, which contradicts reported values (e.g., 84.8 or 82.9). I could not understand what a fairness value of 85 actually means.

3. Evaluation is incomplete. There is no ablation for critical design choices (e.g., the number of sampled models, verifier quality, or extraction accuracy). Moreover, the paper lacks qualitative analysis of failure cases until very late (Figure 2), and even there the discussion remains superficial.

**Questions:**

1. Please provide a precise mathematical definition of fairness and clarify why original datasets do not reach 100 under your metric.

2. Can you provide evidence that multi-model feedback truly outperforms single-model-guided rewrites, beyond performance drop magnitude?At present, I believe that the paired trends in Table 1 alone are not sufficient to draw such a conclusion

3. How sensitive is your framework to incorrect ability extraction or verifier mistakes? Can you quantify this?

---

> ### Author Response · Authors · 2025-12-04
> **Response to Reviewer Mg4w (1/2)**
>
> We thank the reviewer for highlighting key strengths of the paper, including the importance of the benchmark contamination problem, the modularity of the framework, and the novelty of the multi-model feedback design. We respond to the raised concerns below.
>
>
> > W1 & Q1: Evidence that multi-model feedback improves over single-model rewrites
>
> ArenaBencher contains a direct comparison between single-model and multi-model feedback. The configuration with $m = 1$ in our framework corresponds exactly to *single-model* feedback, while $m = 3$ (for $K = 6$ models) is our default multi-model setting. ArenaBencher-1 and ArenaBencher-3 in Table 2 are therefore the single-model and multi-model variants:
>
> - **Difficulty:**  Multi-model feedback consistently produces greater difficulty than the single-model variant. For example:
>   - GSM8K: difficulty increases from 36.3 (m = 1) to 41.4 (m = 3).
>   - CSQA: from 43.3 to 47.0.
>   - Harmful Behaviors: from 13.4 to 24.2.
>   This shows that multi-model feedback finds updates that are harder across the pool, not just for one model.
>
> - **Fairness and alignment:**  Despite higher difficulty, fairness and alignment remain comparable, and sometimes improve:
>   - On Harmful Behaviors, fairness improves from 82.9 (original) and 81.8 (m = 1) to 85.47 (m = 3), while alignment stays above 90.
>   - On GSM8K and CSQA, fairness and alignment for m = 3 are very close to those for m = 1 and remain high overall.
> This indicates that multi-model feedback does not achieve higher difficulty by sacrificing fairness or alignment.
>
> - **Separability:**  Separability under m = 3 remains in a similar range to m = 1. For example, on GSM8K separability goes from 11.3 (m = 1) to 12.2 (m = 3); on Harmful Behaviors from 18.2 to 14.5. Thus, the ability to distinguish models is largely preserved.
>
> Taken together, multi-model feedback yields benchmarks that are:
> 1. Harder across the model pool;
> 2. At least as fair and aligned as the single-model variant;
> 3. With separability that remains comparable.
>
> This justifies our claim that multi-model feedback improves over single-model rewrites in terms of *benchmark quality* as defined by our four metrics, not only in raw performance drop. In the revision, we will (i) explicitly label m = 1 as a single-model baseline in the text and figures, and (ii) devote a short subsection to summarize these comparisons.
>
> We also agree that our wording around “mitigating bias” can be softened. Rather than claiming to remove bias entirely, we will clarify that multi-model feedback *reduces the risk* of strongly model-specific artifacts compared with a single-model objective, and we will point directly to fairness and cross-family trends as supporting evidence.
>
>
>
>
> > W2 & Q2: Fairness: definition and interpretation
>
> Let $ B' $ be the updated benchmark and $ M = \{M_1, \dots, M_K\} $ the model pool. For each model $ M_k $, let $c_k = \sum_{i=1}^{|B'|} \mathbf{1}[\text{FAIL}(M_k, x'_i)]$ denote its total failures on $ B' $, with $ \bar{c} = \frac{1}{K} \sum_k c_k $. We define fairness as $\mathrm{Fairness}(B', M) = \left(1 - \frac{1}{K} \sum_k \frac{|c_k - \bar{c}|}{|B'|}\right) \times 100\%$.
>
> This metric reflects *how evenly failure is distributed*. A score of 100% implies all models fail equally often. In contrast, a score <100 indicates skew, for instance, if weaker models fail much more. This is exactly what we observe on the original benchmarks, e.g., GSM8K fairness of 84.8%, as some models solve far fewer items. ArenaBencher aims to preserve or improve fairness after evolution, avoiding artifacts that make items selectively harder for a subset of models.
>
> We will move this formal definition to Section 4 and clarify that fairness does **not** mean “all models perform equally well,” but rather “failures are not disproportionately concentrated.”

---

> > ### Author Response · Authors · 2025-12-04
> > **Response to Reviewer Mg4w (2/2)**
> >
> > > W3 & Q3: Sensitivity to ability extraction and verifier mistakes
> >
> > Ability extraction and verification are indeed critical components, and we appreciate the opportunity to clarify the results and analysis.
> >
> > Conceptually:
> > - If ability extraction is imperfect but the verifier is correct, misaligned candidates are likely to be rejected by the judge, so they do not enter the final benchmark.
> > - If the verifier itself makes mistakes, it may occasionally accept misaligned or incorrect updates, which can contaminate the benchmark.
> > - Our design tries to minimize this by combining ability-aware prompts with strict verification, then measuring alignment with human evaluation.
> >
> > Empirically, we quantify the combined effect of generator, ability description, and verifier as follows:
> >
> > - **Human evaluation of alignment and correctness.**
> >   On 100 randomly sampled GSM8K updates, three expert annotators assessed:
> >   1. Whether the updated item preserved the original task intent and ability.
> >   2. Whether the question–answer pair was semantically correct.
> >   The results show 95% alignment and 96% correctness, which implies that only 4–5 items out of 100 suffered from misalignment or semantic errors that slipped through the verifier. This gives an upper bound on the effective error rate of the combined ability extraction plus verification pipeline for the final benchmark.
> >
> > - **Failure case analysis.**
> >   Figure 2 shows a representative failure where the verifier accepts an update that subtly changes the required operations. We use such cases to illustrate the residual risk when both ability extraction and the judge fail simultaneously. We agree that the current discussion is brief, and in the revision, we will expand this section with more examples and explicit linkage to the human evaluation results above.
> >
> > To sum up:
> > - Our current quantification comes from human evaluation of the final updates (95% and 96%);
> > - The framework is relatively robust to *moderate* errors because misaligned candidates are filtered by the judge;
> > - Systematic or adversarial failures of both ability extraction and the judge are important limitations and a key direction for future work, for instance, by introducing ensembles or structured verification for critical domains.
> >
> > We hope these clarifications address your concerns about the definition of fairness, the empirical benefits of multi-model feedback, and the robustness of ArenaBencher to imperfections in ability extraction and verification.

---

### Official Review · Reviewer_1g77 · 2025-11-01

**Soundness:** 2
**Presentation:** 3
**Contribution:** 3
**Rating:** 2
**Confidence:** 4

**Summary:**

ArenaBencher is a framework for automatic benchmark evolution that addresses data contamination in LLM evaluation. The system extracts test objectives, generates candidate questions, verifies them with LLM-as-a-judge, and uses multi-model feedback (sampling √K models) to select variants that consistently degrade performance across models. It includes iterative refinement using successful candidates as in-context demonstrations. The framework is evaluated on GSM8K (math), Harmful Behaviors (safety), and CommonsenseQA (reasoning) using 6 open-source models.

**Strengths:**

1. The design avoids creating model-specific adversarial examples that may be trivial for some models, helping ensure that updated benchmarks reflect shared weaknesses across diverse architectures rather than exploiting idiosyncrasies of particular systems.
2. The four desiderata (difficulty, separability, fairness, alignment) provide a principled way to assess benchmark quality.
3. Testing across math, safety, and reasoning domains demonstrates generalizability.
4. 95% alignment and 96% correctness on 100 GSM8K samples provides high alignemnt evidence of the augmented data.

**Weaknesses:**

Though the paper is well-written, there are two fundamental weaknesses I have to bring out:
1. The paper's core motivation is to reduce data contamination's impact on benchmark validity, yet the evaluation metrics do not directly demonstrate reduced contamination or improved prediction of true model capabilities on unseen data by the four metrics (Difficulty, Separability, Fairness, Alignment). The paper needs real-world data validation such as temporal splits or naturally occurring unseen data. For example, GitHub PRs after model training cutoffs provide natural test cases for SWE-bench contamination evaluation (if there is any). The evaluation should test whether ArenaBencher augmentation on augmented SWE-bench better predicts performance on post-cutoff PRs compared to other baselines. Without this validation, it remains unclear whether ArenaBencher actually solves data contamination or merely creates harder variants.
2. The paper provides no comparative evaluation with baseline methods, making it impossible to assess how good the proposed method is. Two critical types of comparisons are missing. First, the paper does not compare multi-model feedback against single-model augmentation to establish cost-effectiveness, leaving unclear whether sampling multiple models provides proportional improvement over using just one model given the additional API costs. Second, there is no comparison with other multi-model collaboration approaches or simpler augmentation methods such as paraphrasing or rule-based perturbations.

**Questions:**

See weakness.

---

> ### Author Response · Authors · 2025-12-04
> **Response to Reviewer 1g77 (1/2)**
>
> We thank the reviewer for highlighting and acknowledging the strengths of our work. Below, we address the two core weaknesses.
>
>
> > W1: Contamination motivation versus evaluation design
>
> We believe that contamination has two related but distinct forms, and ArenaBencher is designed to address both, with a stronger emphasis on the second:
>
> 1. **Explicit contamination (memorization of training corpus).**
>    Benchmark items or very close variants appear directly in the pretraining data, so the model can reproduce answers by memorization.
>
> 2. **Implicit contamination (shortcuts and spurious cues).**
>    The model learns to exploit shallow patterns that are specific to a benchmark’s distribution, such as surface templates, recurring numerical patterns, or common answer formats, without truly acquiring the underlying ability. Performance on the original benchmark can then be inflated even if exact items are not memorized.
>
> Directly measuring explicit contamination is difficult in practice, because (i) the pretraining data of most large models, especially proprietary ones, is not publicly available, and (ii) lexical overlap checks systematically underestimate semantic memorization. ArenaBencher therefore focuses on constructing **new, post-training, evaluation-target-preserving instances** that reduce explicit contamination risk for current models and break benchmark-specific shortcuts:
>
> - **Post-training generation reduces explicit contamination for the evaluated models.**
> All updated test cases in our experiments are synthesized *after* the release and training of the evaluated open source models. These synthetic queries and answers are less likely to have appeared in their pretraining corpora, so explicit memorization of the updated items is largely ruled out for the current model pool.
>
> - **Ability extraction and judge validation mitigate implicit contamination.**
>   Instead of arbitrary adversarial perturbations, ArenaBencher first extracts the core evaluation ability from each original item and then asks the generator and judge to produce new instances that preserve this ability. This design changes surface patterns while keeping the underlying skill fixed. The four metrics then check whether the resulting set is (i) harder, (ii) still able to distinguish models, (iii) fair across models and families, and (iv) aligned with the original task objective. Our experiments show that difficulty increases substantially, while separability and fairness remain high, and human-measured alignment exceeds 95%, which indicates that the updated sets are both harder and still diagnostically useful.
>
>
> In summary, ArenaBencher does not simply generate harder adversarial examples. It constructs post-training, evaluation-target-preserving updates that both remove explicit memorization of the updated items for the current models and weaken benchmark-specific shortcuts, and then verifies that these updates lead to harder yet still fair and aligned benchmarks.

---

> > ### Author Response · Authors · 2025-12-04
> > **Response to Reviewer 1g77 (2/2)**
> >
> > > W2: Missing baselines and comparative evaluation
> >
> >
> > **(a) Multi-model versus single-model feedback.**
> > We included a direct comparison between multi-model feedback and a single-model baseline.
> >
> > - In Section 3.3 we define the scoring mechanism where we sample a subset $M_s \subset M$ of size $m = \lceil \sqrt{K} \rceil$ for feedback.
> > - In Section 4 we explicitly compare two configurations:
> >   1. $m = 1$, which corresponds exactly to *single model* feedback, and
> >   2. $m = \lceil \sqrt{K} \rceil$ (that is $m = 3$ when $K = 6$), which is our default multi model setting.
> >
> > As summarized in Table 1 and Table 2, using multiple feedback models consistently yields stronger benchmark hardening than the single model baseline, where performance drops are larger across tasks, while fairness, separability, and alignment remain similar. This shows that multi-model feedback provides measurable gains over the single model alternative without degrading the other benchmark quality criteria. In the revised version, we will (i) explicitly label $m = 1$ as a *single model baseline* in the text and table captions, and (ii) devote a short subsection in Section 4 to highlight this comparison in a way that is more visible.
> >
> > **(b) Comparison with simple augmentation and other collaborative methods.**
> > We agree that simple augmentation strategies like paraphrasing or rule-based perturbations and other multi-model approaches are useful baselines to discuss. Our main reason for not reimplementing them is the scope, since many such methods are benchmark-specific and focus primarily on surface-level perturbations or single model adversarial prompting.
> >
> > Conceptually, ArenaBencher differs from these approaches in two key ways:
> >
> > - It **explicitly models the evaluation target ability** and uses this to steer generation. Many paraphrasing or perturbation methods apply transformations directly to the original items, which can preserve lexical overlap patterns and do not necessarily produce examples that expose deeper shared weaknesses.
> >
> > - It **optimizes with respect to a multi-model objective** (difficulty plus fairness and separability over a model pool) rather than a single model loss or a single defense. This design is closer to “collaborative evaluation” than to single model adversarial attack frameworks.
> >
> > We hope these clarifications address the reviewer’s concerns and make the contribution of ArenaBencher clearer in terms of both contamination-motivated evaluation and the value of multi-model feedback over simpler baselines.

---

### Official Review · Reviewer_gCrr · 2025-11-03

**Soundness:** 3
**Presentation:** 3
**Contribution:** 2
**Rating:** 6
**Confidence:** 4

**Summary:**

This paper introduces a model-agnostic framework designed to automatically evolve benchmarks to counteract the problem of data contamination in large language model (LLM) pretraining corpora.

The ArenaBencher pipeline operates in four main stages. The authors apply this framework to benchmarks in three domains: mathematical reasoning (GSM8K), commonsense reasoning (CommonsenseQA), and safety (AdvBench Harmful Behaviors).

The results demonstrate that the evolved benchmarks are more difficult for a range of open-source models, improve the separability, and distribute performance drops fairly across the model pool, all while maintaining alignment with the original task objectives as verified by both automated metrics and human evaluation.

**Strengths:**

This paper is easy to follow. It tackles the urgent and widely recognized problem of benchmark contamination and the resulting inflation of LLM performance metrics. I think the proposed direction of creating dynamic, evolving benchmarks is valuable.

The authors have conducted comprehensive experiments to validate their points. The framework is tested across three distinct and important domains (mathematical reasoning, commonsense reasoning, and safety), with LLAMA3, Qwen3, and Mistral.

Difficulty, Separability, Fairness, and Alignment are measured. The inclusion of a human evaluation study to validate alignment and correctness further strengthens the empirical results.

**Weaknesses:**

I think the framework lacks a mechanism to balance the competing goals of difficulty and separability. The results in Table 2 also partly show this. The current selection mechanism, which selects candidates with the highest aggregated loss, may favor test cases that are simply too hard for all models, causing their scores to cluster near zero and reducing the benchmark's ability to distinguish between them. A truly discriminative benchmark should maximize the variance in performance across models.

The design choice of sampling $m = \lceil\sqrt{K}\rceil$ models for feedback should be better justified. Although the paper cites "classical ensemble heuristics" from Random Forests and XGBoost, ArenaBencher's objective is different. There is no guarantee that a heuristic designed for decorrelating learners by subsampling features would be optimal for estimating a population statistic by subsampling models.

The entire framework is dependent on a single proprietary model (GPT-4). The authors commendably include a failure case analysis for the generator (Figure 2), but do not discuss or analyze the robustness of the verifier/judge. A systematic bias or blind spot in the judge model could silently poison the entire evolved benchmark.

**Questions:**

While the related work section effectively contrasts the approach with single-model adversarial attacks, it could be improved by including more literature on dynamic benchmark generation.

The authors could provide ablation studies or theoretical arguments to support the $m = \lceil\sqrt{K}\rceil$ choice over alternatives.

Could the authors provide an analysis of the computational cost of this process? How does the cost scale with the benchmark size $N$, the model pool size $K$, and the number of refinement rounds $R$?

---

> ### Author Response · Authors · 2025-12-04
> **Response to Reviewer gCrr**
>
> We thank the reviewer for the thoughtful and detailed comments, and for recognizing the importance of dynamic benchmark evolution and our use of multi-model feedback and human evaluation. Below, we address the main concerns in turn.
>
> > W1: Difficulty versus separability
>
> Our primary design goal is to increase difficulty while preserving, rather than maximizing, separability. The current selection mechanism therefore optimizes for difficulty, and we evaluate the resulting benchmarks using four metrics, including separability. As shown in Table 2, difficulty increases substantially across all three domains, while separability remains in a comparable range and does not collapse toward zero. For example, separability changes from 15.2 to 12.2 on GSM8K and from 8.5 to 7.2 on CSQA, indicating that the benchmark still spreads model performance and retains discriminative power even under higher difficulty. At the same time, fairness improves or stays similar, which suggests that the difficulty increase does not come from adversarial artifacts that selectively target specific models.
>
> We agree that explicitly trading off difficulty and separability at selection time is an interesting extension. We plan to include a discussion of simple variants such as selecting candidates within a target accuracy band or using a composite score that combines loss with cross-model variance. These designs would more directly optimize for a balanced operating point while keeping the core ArenaBencher framework unchanged.
>
>
> > W2: Choice of subsampled models for feedback
>
> Our objective is to approximate the expected loss of a candidate under the model pool while keeping the cost tractable. Averaging losses over a random subset of models provides an unbiased estimator of the full pool average, with variance that decreases as $m$ increases. This creates a natural bias-variance tradeoff that larger $m$ reduces estimator variance, but increases computational cost.
>
> We chose $m = \lceil \sqrt{K} \rceil$ as a simple rule that keeps per-query cost sublinear in $K$ while still providing diverse feedback. Empirically, we compare $m = 1$ and $m = 3$ (which equals $\lceil \sqrt{6} \rceil$ in our six-model pool). As shown in Table 1 and Table 2:
> - $m = 3$ consistently yields larger drops in accuracy and larger increases in ASR than $m = 1$, indicating more effective benchmark hardening, and
> - fairness and alignment remain high and similar between $m = 1$ and $m = 3$, while separability stays in a similar range.
>
> These results suggest that using a small but diverse subset of models improves the quality of updates compared with a single model, while avoiding the cost of evaluating all models on every candidate. We will make this tradeoff and the unbiased estimation view more explicit and move the $m$ ablation into the main text for clarity.
>
>
>
> > W3:  Dependence on a single proprietary judge
>
> We agree that the reliability of the judge is critical. We took two steps in the current work.
> **First**, we empirically validate the judge through human annotation. For 100 randomly sampled GSM8K updates, three expert annotators evaluated each item for (i) alignment with the original test objective and (ii) correctness of the question answer pair (Lines 427 - 457). 95% of the items were judged aligned, and 96% were judged correct, which indicates that the judge is reliable in the vast majority of cases.
> **Second**, we explicitly analyze failure modes. Figure 2 presents a case where the judge incorrectly accepts an update that introduces a missing constraint and a shift in required operations. We highlight these failure modes as limitations of the current reliance on a single judge and discuss them in the conclusion.
> We agree that further strengthening this component is important future work. We will (i) clarify that the generator and judge are logically separated modules, and (ii) add a discussion of using ensembles of heterogeneous judges or structure-aware constraints as additional safeguards, which is orthogonal to the core multi-model feedback mechanism.

---

> > ### Author Response · Authors · 2025-12-04
> > **Response to Reviewer gCrr (2/2)**
> >
> > > Q1: Additional related work on dynamic benchmark generation
> >
> > Thank you for the suggestion. We cite several dynamic or live benchmark efforts, but we agree that this discussion can be clearer and more complete. We have expanded the subsection of related work on benchmark updating to more systematically cover recent dynamic benchmark generation. So that we position ArenaBencher as complementary in a way that our focus is on ability preserving and multi-model feedback-driven evolution of existing benchmarks, rather than proposing a new task suite from scratch.
> >
> >
> >
> > > Q2: Ablation and theoretical support
> >
> > From a theoretical perspective, the estimator $\hat{L}(x) = \frac{1}{m} \sum_{k \in S} L_k(x)$ is an unbiased estimator of $L_{\text{full}}(x)$ whenever $S$ is a uniformly sampled subset of size $m$. Its variance scales approximately as $\operatorname{Var}[\hat{L}(x)] \propto 1/m$ for fixed $K$ and candidate $x$. Increasing $m$ therefore reduces the noise in the estimated loss but increases the inference cost. This is exactly the tradeoff we need to manage, since the total number of evaluations scales as $O(N \cdot R \cdot n \cdot m)$, where $N$ is the benchmark size, $R$ is the number of refinement rounds, and $n$ is the number of candidates per seed item.
> >
> > Empirically, we provided an ablation on $m$ by comparing $m = 1$ and $m = 3$ (with $K = 6$, so $m = 3$ corresponds to $\sqrt{K}$). As reported in Table 1 and Table 2:
> > - $m = 3$ consistently yields larger accuracy drops and higher ASR than $m = 1$, indicating more effective benchmark hardening, and
> > - fairness, alignment, and separability remain similar between the two settings.
> >
> > These results suggest that using a small subset with $m > 1$ improves the quality of updates over a single-model estimator, without degrading other desirable properties of the evolved benchmark.
> >
> >
> >
> > > Q3: Computational cost and scaling
> >
> > The dominant cost of ArenaBencher comes from model inference on the generated candidates. For each original test instance, we perform the following operations per refinement round:
> >
> > 1. Generation and judging.
> >    For each seed item, the generator produces $n$ candidate variants, and the judge filters them. This part uses only the generator and judge model and therefore has a cost that scales as $O(N \cdot R \cdot n),$ independent of the model pool size $K$.
> >
> > 2. Pool evaluation with subsampled models.
> >    For each surviving candidate, we evaluate its loss using a subsampled set of $m = \lceil \sqrt{K} \rceil$ models from the pool. This yields a total of $O(N \cdot R \cdot n \cdot m)$ evaluations on the target model pool. Since $m$ grows sublinearly in $K$, this keeps the per candidate cost sublinear in the pool size while still capturing diversity across model families.
> >
> > In practice, we keep $R$, $n$, and $m$ as small constants in all experiments. For example, in our GSM8K and CSQA runs, we use $R = 3$ refinement rounds, $n = 5$ candidates per seed item, and $m = 3$ models per candidate (with $K = 6$). We also subsample the original benchmark for evolution. Under these choices, the total number of model evaluations is on the order of a few times the original benchmark size multiplied by small constant factors, which keeps computation manageable on a single multi-GPU node.
> >
> > We will add this scaling analysis and a short paragraph with concrete numbers for GSM8K and CSQA in the revised version to clarify the practical cost of applying ArenaBencher.

---

### Meta-Review · Area_Chair_nJZy · 2026-01-07

**Summary:**

This paper introduces an automatic framework (ArenaBencher) for benchmark evolution that aims to mitigate data contamination and benchmark saturation by iteratively generating and selecting updated test cases using multi-model competitive feedback. The work is timely and well-motivated, given growing concerns about benchmark leakage and overstated progress in foundation models. Reviewers generally agree that automatic benchmark evolution is an important direction and that ArenaBencher shows promising empirical results. However, reviewers also identify several important issues that limit the strength of the current submission.

A primary concern is whether the evaluation convincingly demonstrates that ArenaBencher addresses data contamination rather than merely producing harder benchmark variants. While the paper introduces four  metrics (difficulty, separability, fairness, alignment), reviewers note that these do not directly validate reduced contamination or improved prediction of true generalization on unseen data. The rebuttal clarifies the distinction between explicit and implicit contamination and provides a principled justification for the chosen metrics, but the lack of real-world or temporal validation remains a limitation.

Reviewers also question several design choices and methodological assumptions. These include the selection mechanism to balance the difficulty and separability, the choice of subsampling models for feedback, and the heavy dependence on a single proprietary judge model. While the rebuttal provides theoretical justification, ablation results, and human validation of the judge, concerns remain about robustness to judge bias and the absence of more extensive ablations or alternative verifier designs.

Finally, reviewers point out gaps in baseline comparisons and empirical completeness. Although the paper includes comparisons between single-model and multi-model feedback within ArenaBencher, it lacks direct comparisons to simpler augmentation strategies or other multi-model benchmark evolution approaches. Several reviewers also note that metric definitions (e.g., fairness) and failure analyses were initially unclear, though these are clarified in the rebuttal.

Overall, ArenaBencher presents a useful framework with good motivation, but reviewers agree that stronger empirical validation, particularly demonstrating reduced contamination and improved generalization, and clearer justification of design choices would be needed to support acceptance at ICLR.

**Reviewer Concerns:**

I believe that most reviewers’ core concerns remain partially outstanding, despite a thoughtful and substantive rebuttal.

Reviewer gCrr’s concerns regarding the difficulty–separability trade-off, justification for subsampled multi-model feedback, dependence on a single judge, and computational cost are partially addressed through theoretical explanations, ablations, and human validation. However, explicit mechanisms for balancing separability and difficulty and stronger robustness analysis of the judge remain open.

Reviewer 1g77’s concerns about the lack of direct evidence that ArenaBencher mitigates contamination, and the absence of strong comparative baselines, remain largely unresolved. While the rebuttal provides conceptual clarification and internal comparisons, empirical validation on truly unseen or post-cutoff data is still missing.

Reviewer Mg4w’s concerns about unclear metric definitions, limited ablations, and insufficient evidence supporting claims about bias mitigation are addressed in part by formalizing fairness and clarifying single- vs. multi-model comparisons. However, sensitivity to ability extraction and verifier errors remains only partially explored.

**Reviewer Scores:**

A full discussion would likely result in small positive score adjustments, driven by the rebuttal’s clarifications, added theoretical justification, and human validation of the judge. However, key concerns regarding contamination validation, robustness, and baseline completeness, are not fully resolved and would require substantive additional experiments beyond rebuttal alone. As a result, overall reviewer sentiment would likely remain cautious.

---

### Decision · Program_Chairs · 2026-01-26

Reject